# The Impact of Organic Traffic of Crowdsourcing Platforms on Airlines' Website Traffic and User Engagement

**Damianos P. Sakas and Dimitrios P. Reklitis ***

Department of Agribusiness and Supply Chain Management, School of Applied Economics and Social Sciences, Agricultural University of Athens, 1st km Old National Road Thiva–Eleusis, 32200 Thiva, Greece; d.sakas@aua.gr
* Correspondence: drekleitis@aua.gr; Tel.: +30-698-801-9428

**Abstract:** With airline companies increasingly relying on crowdsourcing websites to deploy their digital marketing strategies, marketeers and strategists seek to acquire an understanding of the factors affecting airlines' organic traffic and user engagement. Such an understanding is acquired through the consideration of variables that influence a company's organic traffic and user engagement and their correlation to each other. A three-stage data-driven analysis is used to examine the correlation between the foregoing variables and to consider strategies that can be implemented to optimize organic traffic and user engagement. The first section gathers data from five airline companies' websites and five crowdsourcing websites over an interval of 180 days. The second stage creates an exploratory diagnostic model, through Fuzzy Cognitive Mapping, to visually illustrate the cause-and-effect correlations between the examined metrics. Finally, a predictive micro-level agent-based model simulates optimization strategies that can be used to improve organic traffic and user engagement. The results of this study, reveal that crowdsourcing organic traffic increases airline websites' user engagement through paid campaigns, while a limited correlation was found to exist between the average duration of a user to organic traffic. The results of this study provide tangible digital marketing strategies which can be used by airline companies to improve the influence of their digital marketing strategies on their users.

**Keywords:** crowdsourcing; big data; web analytics; SEM; strategy; user engagement; crowd branding; airlines; web traffic; digital marketing; brand name





## 1. Introduction

Crowdsourcing consists of a combination of "outsourcing" and "knowledge of the crowds". In essence, crowdsourcing is a process wherein tasks are provided to the public via an open call [1,2]. Thus, in a crowdsourcing project, each crowd member with particular knowledge of a given crowd can be assigned to complete an element of a project for which they will be compensated. This method of project completion is an attractive model for enterprises and crowd workers alike [3], as it allows a service model to be adapted to the characteristics and requirements of a given crowd. For instance, businesses have increasingly reconfigured their delivery service models based on novel consumer demands and progressively cater to ridesharing same-day delivery services. The ability of crowdsourcing to adapt business models to crowd demands is known as crowdsourcing logistics [1,4]. Although not yet widely utilized in all sectors, crowdsourcing is a promising new project model for business enterprises, for example, urban logistics [5] and manufacturing [6]. Indeed, the increasing popularity of crowdsourcing, which relies on computers to complete most of the assigned tasks, has created some of the most renowned crowdsourcing platforms and service providers [7,8].

Crowdsourcing platforms are used to host groups of people that answer to specific challenges or tasks by providing ideas, perceptions or business plans for new and innovative services or products [6,9]. In this article, five crowdsourcing platforms are being used for the purpose of the study. On the other hand, crowdfunding platforms connect

businessmen or companies with potential customers. With the help of these platforms, entrepreneurs gather funds, while they promise some type of reward when the service or the product reaches the market [10].

Usually, crowdsourcing projects have a goal for the benefit of a community, the public and digital marketing play a crucial role in that. Digital marketing helps to make people aware of an issue that needs to be solved [11].

### 1.1. Participatory Culture and Mind Sharing

Two core aspects of crowdsourcing projects are participatory culture and mind sharing. An important benefit obtained when relying on crowdsourcing knowledge is access to objective decision-making skills [9]. Access to crowd knowledge has been termed as "mind sharing". By relying on a crowd to inform decision-making, the individual components which constitute the given problem are weighed objectively and, thus, the decision-making process remains free from personal biases which may exist [12]. The argument is that through mind sharing, decision making can be improved in terms of efficacy, timeliness and objectiveness [9,12]. Managers can rely on crowdsourcing practices to generate ideas and make decisions by using mind sharing because the main goal of crowdsourcing is to make people "work *with* us" and not just "work *for* us" [9,13].

Another element of mind sharing in crowdsourcing is the independent thinking factor, which essentially means that there is no right or wrong decision or opinion, and the focus is on opinion diversity [9,14]. Mind sharing in crowdsourcing is different from group thinking [9] because group thinking has a limitation—conformity. Group thinking can produce faulty decisions because participants prioritize group harmony over practical, truthful resolutions of an issue [14].

In order to achieve mind sharing, it is imperative to create a participatory culture. Contrary to consumer culture, participatory culture motivates individuals to act as contributors and not just as consumers [15,16]. At first, an individual does not need to be part of a specific crowd, and through participatory culture, the individual becomes part of the crowd; that's why participatory culture is crucial in crowdsourcing and crowdfunding [17,18]. Crowdsourcing happens through participation and the need to participate [15]. Crowdsourcing and crowdfunding projects cannot be productive without the implementation of the mindset of participatory culture [18]. This is because the goal of crowdsourcing is to gather ideas and solutions, while the goal of crowdfunding is to gather investors and funding from a large group of people [6].

As reliance on digital data becomes increasingly popular, so does the popularity of "Open Data". Open Data is a philosophy that maintains that some types of data should be freely available to the public for publication and copying without any limitations being imposed by copyrights, patents or other forms of regulatory mechanisms [19]. The increasing reliance on digital technologies has not only created implications for Open Data but has also led to the creation of "Big Data". The term "Big Data" refers to an existing volume of data that exceeds the efficient storing, managerial and processing abilities of existing forms of technologies, and through Big Data analysis, companies can achieve a competitive advantage [20].

Due to the rise of Big Data, a process that could measure, select and analyze large amounts of information had to be created. This process, known as "Web Analytics", is used with a view of understanding and improving web usage. Various studies presented the impact of Web Analytics on both the logistics and the crowdsourcing sectors [21,22]. This type of examination has allowed for the measurement of various variables, which, amongst others, include Web Traffic. Web Traffic is a variable which transcribes the volume of data that a given user sends to a website from their web browser. Web Traffic is measured by analyzing a combination of the number of visitors (traffic) and the number of pages visited. It should also be noted that the measurement of Web Traffic is impacted by the user's keyword density, that is, the keywords that have been searched by the user prior to accessing the intended site.

As users visit a company's website, they "deposit" large amounts of data which, holistically, constitute a large Big Data pool that can, thereafter, be utilized by marketeers and strategists to conceptualize and structure their marketing strategies more efficiently. Indeed, factors, such as the structure of a digital marketing strategy and web visitors' capabilities, significantly influence the level of interaction between a user and a component of the digital site. In this capacity, crowdsourcing platforms offer services that can substantially improve a company's brand by increasing the company's visibility and popularity on digital platforms [23]. An increase within the digital platform's visibility generates larger amounts of Web Traffic which, in turn, allows the site to rank higher in terms of SEO and SEM. Incidentally, higher ranking and larger amounts of Web Traffic can be used by marketeers to improve their company's brand through the deployment of tailored digital marketing strategies.

This study relies on Web Analytic metrics to narrow the examination between the correlations of crowdsourcing platforms and logistics websites. In particular, this paper analyzes the User Engagement and Global Rank metrics of selected airline companies, as gathered through SEMRush, and also those provided by selected crowdsourcing websites. The results of this study offer the possibility of identifying the impact that crowdsourcing websites have on a given airline company's brand. From a philosophical point of view, this type of data collection would be on the outskirts of the term "crowdsourcing", and for this reason, the data collected from web analytics can be considered passive crowdsourcing [24].

### 1.2. Impact of Crowdsourcing on Big Data and User Engagement

Although crowdsourcing is not solely a digital activity, the most effective manner of interacting with the crowd is through the online medium, wherein crowdsourcing platforms act as "conduits" between clients and the crowd [1,3]. The line that connects the aforementioned digital activity and the results is the analysis of the Big Data.

Digital entities and companies can acquire a competitive advantage through the analysis and structuring of their Big Data [25]. By the term "Big Data", researchers describe the abundance of data in an unstructured form. In order to acquire insights from these vast amounts of data, there is a need to structure and direct the data to the "right path" [26]. Several studies have revealed that numerous industries can benefit from the efficient use of Big Data; such industries range from humanitarian actions to government decision-making and crowdsourcing [27–29]. According to Zheng et al., "*crowdsourcing is a process of acquisition, integration, and analysis of big and heterogeneous data generated by a diversity of sources in urban spaces, such as sensors, devices, vehicles, buildings and human*" [29].

The availability of a wide pool of experts offered by crowdsourcing is especially beneficial to businesses with a more restricted talent pool [30]. Nevertheless, the affordability of crowdsourcing platforms coupled with access to open data and a wide talent pool represent key motivators for small and large businesses alike to rely on crowdsourcing platforms for their production of goods and services.

The conception of crowdsourcing is based on the notion that the rise of digital technology facilitates the sharing of ideas and efforts worldwide [31]. Thus, the functioning of crowdsourcing platforms is designed in a manner that ensures the constant transfer of Big Data between collaborating businesses throughout the entire duration of their cooperation.

In problem-solving projects, which are the core focus of crowdsourcing platforms, user engagement plays a crucial role, wherein user engagement represents the preservation of various social connections with a great level of participation [32]. The term "user engagement" refers to the understanding of the user's "emotional engagement", that is, the user's feelings in given situations, such as love and death, as well as the understanding of the "cognitive engagement", which explains ordinary events [33,34]. According to de Vreede et al., user engagement in crowdsourcing has three crucial parameters; motivation to contribute, goal clarity and personal interest in the subject [34]. Applying this to the airline industry, it will be critical for airline companies to consider what attracts and engages their

users in order to produce valuable contributions to an issue, such as creating an airline company's logo [35].

Crowdsourcing websites offer the opportunity for airline companies to begin exploring the parameters which influence user engagement and implicit participation, as crowdsourcing web pages and airline pages share a "connection". In essence, airline companies can post banners or projects which concern their company on crowdsourcing platforms [35]. When a user enters a crowdsourcing platform's webpage, they are able to immediately view the banner or project which relates to the airline company's website. Once the user selects the banner appearing on the crowdsourcing platform, they are automatically redirected to the airline company's website. This process of redirection of users from crowdsourcing platforms to airline company's websites creates added value to the airline company's brand name because, through a redirect, the airline company's website generates additional traffic, which, in turn, influences the airline company's Global Rank [26].

In addition to user engagement, another crucial aspect of crowdsourcing is participatory culture. Participatory culture includes implicit and explicit participation [36]. In the past, participation was seen as an act. For instance, the act of the website user creating a comment amounted to participation. This type of participation constitutes explicit participation [37], which is determined by motivation. Some examples of explicit participation include activism and writing on a blog [37]. Unlike explicit participation, implicit participation does not require conscious activity [37]. According to Schäfer, "*in information management systems, participation rather unfolds implicitly, and many users are actually not aware that they contribute to an application simply through using it*" [37]. Some examples of implicit participation include using rating platforms and accessing websites [37].

This study focuses on implicit participation by relying on metrics that contribute to accessing websites, such as Average Visits Duration (AVD). AVD is not only a user engagement metric [38] but also an implicit participation metric, as it provides insights to crowdsourcing platforms [37]. Finally, all the research hypotheses put forth in this paper focus both on user engagement and implicit participation, as participation is a crucial aspect of crowdsourcing.

### 1.3. Research Hypotheses

Due to the competitive environment in which large airline companies operate, it is crucial for them to possess a clear understanding of the factors influencing their organic traffic and user engagement. "Organic traffic" refers to the number of visitors who visit a website from an unpaid source [38]. The user engagement metric is constituted by a number of metrics, including the "Average Visit Duration", the "Pages per Visits", and the "Unique Visitors" metrics [39,40]. It is especially important for such organizations to comprehend the efficiency of their digital organic traffic, as this variable has considerable influence over user engagement, with the opposite being equally true. In recent years, the airline sector has increasingly relied on digital technologies such as crowdsourcing platforms to execute its corporate tasks. Due to the competitive nature of the airline industry and the growing reliance on digital technologies by airline companies, it has become increasingly important to examine the impact that the outsourcing of projects to crowdsourcing platforms has on an airline company's user engagement and website traffic.

The present research addresses this emerging field by examining the variables which influence organic traffic and user engagement and further considers the web metric variables which are most influenced by variations in the user engagement of a given airline website. The insights obtained from the outcome of such research would provide valuable feedback on the efficiency of airline companies' website activities. In particular, the insights would allow:

1.  Decision makers to better understand the impact of outsourcing projects to crowdsourcing platforms. This can be accomplished by giving business planners the ability to visualize how various online metrics affect Organic Traffic and User Engagement.

There are numerous parameters that affect Organic Traffic, such as brand name and which decision makers should consider [41].

2.  Web Developers to acquire a holistic understanding of the impact of their company's Organic Traffic and User Engagement. By considering the influence which web analytics and web metrics have on corporate operations, web developers may be able to consider strategies that can be implemented to maximize company profits. Metrics that can be of use to web developers include CSS metrics. As such, through the understanding of web metrics and the consideration of web analytics, web developers may be able to contribute to corporate strategies.

3.  Marketeers to gather crowdsourcing data which can be used to restructure their digital marketing strategies based on the influence of web metrics variables. Digital marketing is useful for crowdsourcing platforms as digital marketing campaigns and advertising enable the identification of problems and areas in need of development [11]. For instance, an organization would be able to utilize variables, such as "traffic paid costs" or "paid keywords costs", in order to increase User Engagement on their digital platform.

With a view of examining the influence that Big Data (extracted from crowdsourcing platforms) has on an airline company's Big Data (as found on its website), the present paper considers the following research hypotheses:

**Hypotheses 1 (H1).** *The "Organic Traffic" of airline companies affects the "Paid Traffic Cost" variable of airline company's websites through their "Unique Visitors" metric.*

This hypothesis seeks to identify whether the "Unique Visitors" as a parameter of User Engagement affects the "Paid Traffic Cost" and influences the organic traffic found on airline companies' platforms in order to identify the possible intercorrelations between the gathered metrics.

The examination of the proposed hypothesis H1 seeks to identify the tangible benefits which airline companies acquire through the delegation of projects to crowdsourcing platforms. This research hypothesis may result in the creation of "crowd branding", referring to the use of a group of people in the creation and communication of brand identity and brand image [42]. Thus, the outcome of this research hypothesis may reveal the impact that crowd branding has on the creation and establishment of a brand.

**Hypotheses 2 (H2).** *"Organic Traffic" of crowdsourcing platforms affects the "Paid Keywords" of airline companies through their "Unique Visitors" metric.*

This hypothesis seeks to identify whether the "Unique Visitors" as a parameter of User Engagement affects the "Paid Keywords" and influences the organic traffic found on crowdsourcing platforms in order to identify the possible intercorrelations between the gathered metrics.

**Hypotheses 3 (H3).** *The "Organic Traffic" found on crowdsourcing platforms influences the "Organic Traffic" of airline companies as a result of the airline companies' "Average Visits Duration" variable.*

This hypothesis seeks to identify any potential correlations which may exist between the organic traffic found on crowdsourcing platforms and that found on airline companies' websites and how those two are affected by "Average Visits Duration".

**Hypotheses 4 (H4).** *The "Average Visits Duration" metric impacts the "Organic Traffic" of airline companies and influences the "Global Rank" metric of airline companies' websites.*

This research hypothesis allows for the examination of a potential correlation between "Average Visits Duration" and User Engagement parameters, such as general visits, average

visits duration and pages per visit with the "Global Rank". The identification of such a correlation or a lack thereof would provide marketeers with considerable insights, which are expected to assist in the identification of valuable advertising investments for the airline company.

**Hypotheses 5 (H5).** *The "User Engagement" metric affects the "Global Rank" of airline companies through the "Total Crowdsourcing Users".*

This hypothesis seeks to identify whether all the User Engagement variables combined (Average Visits Duration, Pages per Visits, Unique Visitors), represented by the "User Engagement" metric, influence the "Global Rank" through the "Total Crowdsourcing Users" as presented from the Deloitte research [43]. This hypothesis considers whether it is beneficial for airline companies to rely on and make use of crowdsourcing platforms.

**Hypotheses 6 (H6).** *Paid Keywords and Paid Traffic Cost of airline companies affect the Organic Traffic of the selected airline companies.*

One of the principal objectives of this study is to elucidate the implicit participation procedure of potential website users. In order for potential users to visit a website that does not possess a significant digital brand name, they follow directed visits through Paid Traffic Costs and Paid Key Words. Through this process, and once the user has been convinced to visit the airline company's website, the quality of the company is established in the user's mind. Thereafter, the user will visit the airline company's website through Organic Traffic. Therefore, the question remains; is a user's participation influenced and stimulated by the paid campaigns of the corporate website?

The research hypotheses H1, H2, H3, H4 and H6 include characteristics of implicit participation, as they embrace elements of the procedure required to access a website. Such elements include the manner in which users access the website (Organic Traffic) and the method used to access it (Paid Traffic Cost).

## 2. Methodology

With airline companies increasingly relying on crowdsourcing websites to deploy their digital marketing strategies, marketeers and strategists seek to acquire an understanding of the factors affecting airlines' Organic Traffic and User Engagement. This study adopts a pioneering methodology in order to pursue a different strategic perspective for understanding a company's brand name and user engagement through its web metrics. In addition, this paper illustrates, analyzes and simulates the above metrics and the correlations between them. The exploration of this problem relies on the deployment of the following three-step methodological process: (a) the collection and statistical analysis of Big Data from airline companies and crowdsourcing platform websites, (b) the visual representation of existing correlations through the use of Fuzzy Cognitive Maps and (c) the use of an agent-based model (ABM) for the construction a predictive user-engagement simulation.

### 2.1. Collection and Statistical Analysis of Big Data

Through the use of SEMRush API, Big Data generated from 5 airline companies (Emirates International Airline, Qatar Airways, Turkish Airways, AirFrance and Lufthansa) and 5 crowdsourcing platforms (Indiegogo, Microworkers, Upwork, Amazon MTurk, Patreon) was retrieved over a continuous time interval of 180 days, in order to examine the varying fluctuations and correlations which exist amongst the examined metrics (Organic Traffic, AVD, Unique Visitors, Paid Keywords, Paid Traffic Cost, Pages Per Visit, Global Rank). The selected airline companies examined within this study were chosen based on (a) their industry popularity and (b) their geographic coverage, while the crowdsourcing platforms examined were selected based on (a) their crowd size, (b) their crowd interaction and (c) their crowd expertise in the airline industry.

In regard to the crowdsourcing websites which are used in the scope of this research, it is important to note that crowdfunding platforms have also been used for this study. These platforms have been selected in order to identify the participation which occurs in each crowdsourcing or crowdfunding project. This study examines implicit participation but also considers explicit participation, which can be identified through the number of discussions, wherein users submit questions. In most cases, product development companies interact with the crowd by responding to the questions submitted [44]. In order to enhance participation in the crowdsourcing platform, the user is given the ability to react through "like" or "dislike" options. This strategy can be observed on the Patreon website; in order to increase user engagement and participation, Patreon has deployed a core feature that allows the website to interact with communities [45].

The statistical analysis of the Big Data samples sought to assess the extent of the influence of organic search traffic on User Engagement within the selected websites. Furthermore, the statistical analysis identified the existence of several correlations between the examined metrics. The identification of such correlations was executed using the Pearson $\rho$ coefficient. The metrics used in this study are illustrated in Table 1.

**Table 1.** Description of the web metrics.

| Metric | Description |
|---|---|
| Organic Traffic | The metric referred to as "organic traffic" refers to the number of visitors who visit a website from an unpaid source (i.e., another website or search engine). This metric is gathered from web analytic platforms, and, as such, it cannot be decomposed into different variables [46]. |
| Paid Keywords | A keyword is the sentence, or a single word used to enter in search engines by users wanting to access a website to find something specific [47]. "Paid keywords are keywords you bid for inside Google Ads" [48]. |
| Paid Traffic Cost | "Paid Traffic cost estimates the total amount spent on attracting traffic" [40]. |
| Average Visits Duration | Average Visits Duration is defined as "an average estimate of the amount of time spent on the site during each visit" [49]. Average Visit Duration is a metric that has been gathered from SEMRush and is termed "Average Visit Duration". Given that this paper has relied on SEMRush to obtain the data, similar terminology has been used throughout the study for the avoidance of confusion. |
| Unique Visitors | "An estimate of total unique visits to the website over the chosen month" [41]. |
| User Engagement | Constituted by the "Average Visit Duration", the "Pages per Visits" and the "Unique Visitors" variables [38,41,50,51] |
| Global Rank | "Global Rank" is a ranking system of the domain names appearing on a web browser. The ranking is based on the organic visibility of a given domain. Organic visibility itself is based on the monthly organic traffic which a given domain generates from organic searches [41]. |

### 2.2. Visualizing Correlations Using Fuzzy Cognitive Maps

The collection of Big Data from the examined websites and their subsequent statistical analysis revealed the existence of correlations between the examined metrics. The strength of these correlations was presented in a visual form using Fuzzy Cognitive Maps, which can be applied within marketing strategy and decision making [52]. The main objective of implementing this model is to provide a visual representation of the correlations, both positive and negative, and the relationship between the examined metrics [53]. In addition, as shown in Figure 1, the use of Fuzzy Cognitive Mapping allows for the identification of positive and/or negative correlations between the analyzed metrics.

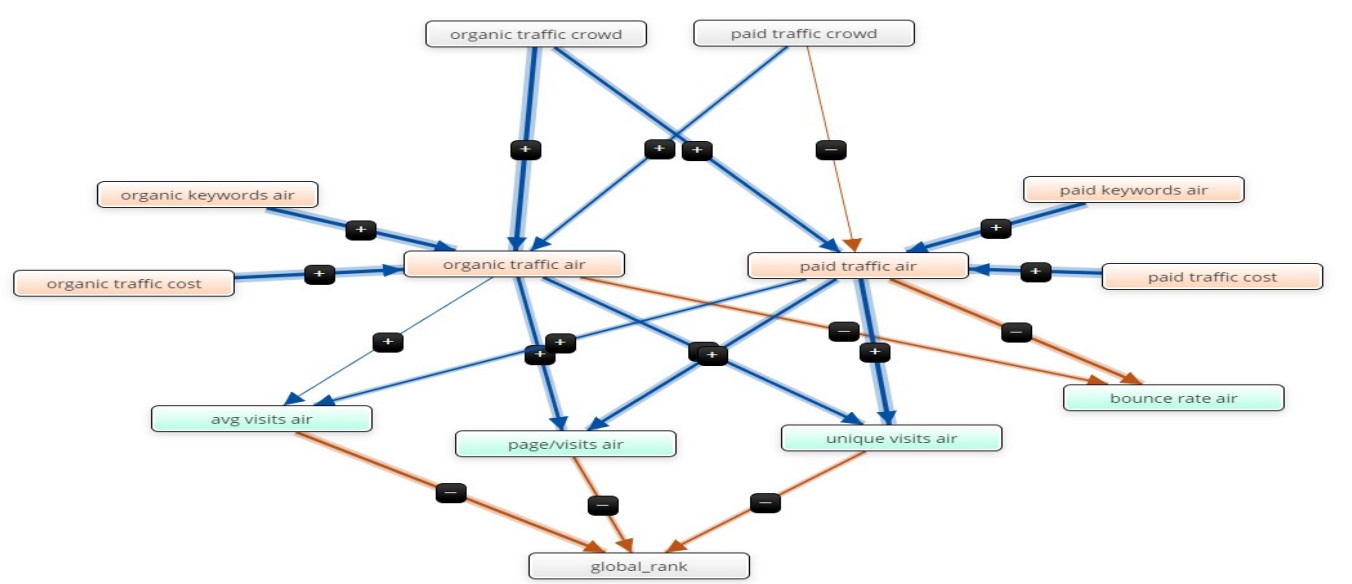

**Figure 1.** Fuzzy Cognitive Map used to present the correlations between the gathered metrics of airlines and crowdsourcing companies. The Fuzzy Cognitive Map has been created through Mental Modeler (mentalmodeler.com, accessed on 14 June 2021).

The output of the statistical analysis further revealed the existence of several variations, namely, in terms of the organic traffic of crowdsourcing platforms and the paid keywords of airline websites, as well as variations between the User Engagement metrics and, for example, pages per visit and unique visitors.

### 2.3. Implementation of a Predictive Agent-Based Model

Agent-based models (ABMs) operate through the interaction of several features, which are defined as agents [54]. Within ABMs, agents are characterized by certain attributes and operate within a system based on prescribed "if–then" rules [54]. ABMs have a dual emphasis—considering both agent behavior within the system and system operations as a whole [55]. In essence, an ABM observes, first, branded search traffic and generates web behavioral data on the user engagement on the website [41]. Second, an ABM provides feedback on the different interactions which occur within the website [41].

The output of the statistical analysis and the visual representation of the identified correlations and variations between the examined metrics were adapted to an ABM in order to simulate future User Engagement and Global Rank within the selected websites. The implementation duration of the ABM lasted 220 days. This process is supported by previous studies which adopted a similar methodological framework [56,57].

The adoption of agent-based modeling is especially beneficial to marketeers as the simulation provided by the model allows for the development of effective digital marketing strategies which are based on predicted User Engagement and Global Rank [58]. Companies cooperate with a large number of crowdsourcing platforms that generate traffic affecting their brand name and user engagement. By using the data produced by these crowdsourcing platforms, marketeers will be able to create a more effective digital marketing strategy. Conversely, the use of this data will enable marketeers to predict the brand name, user engagement and the profit of their website without incurring significant marketing costs.

### 3. Results and Analysis

This section displays the results generated from the data gathering platforms, such as SEMRush. The results in question consist of raw data (collected from airline companies'

websites and crowdsourcing websites) which have been analyzed using statistical analysis tools, as displayed in Table 1.

Following the collection of data from the examined airline companies' websites and from crowdsourcing websites, the data of each group were combined in order to carry out sector-specific data analysis (per sector). Table 2 depicts the chosen descriptive statistics such as mean, min, max and standard deviation for each of the web metrics. These statistics are the result of gathered web metrics over a continuous period of 180 days. For example, the "Organic Traffic Crowdsourcing" standard deviation is the deviation of all five crowdsourcing companies over an interval of 180 days, as described in Section 2.1. The mean of the "Organic Traffic Airlines" refers to the mean result of five gathered airline companies over a continuous period of 180 days.

**Table 2.** Descriptive statistics illustrate the relevant metrics of airline companies and crowdsourcing companies' websites within the previous 180 days period.

| | Mean | Min | Max | Std. Deviation |
|---|---|---|---|---|
| Organic Traffic Crowdsourcing | 24,986,409.66 | 20,559,872.00 | 32,313,788.00 | 922,337.20 |
| Organic Traffic Airlines | 372,295,511.50 | 676,850.00 | 1,219,572,795 | 533,484,436.09 |
| Average Visits Duration | 2496 | 2422 | 2626 | 74.52516 |
| Unique Visitors | 24,360,059.16 | 22,590,969 | 26,468,516 | 1,605,213.73 |
| Paid Keywords | 226,309,525.16 | 9223.37 | 742,321,771.00 | 307,516,792.67 |
| Paid Traffic Cost | 222,409,990.66 | 9223.37 | 713,699,149.00 | 296,088,133.17 |
| Pages Per Visits | 20.68 | 20.04 | 21.70 | 0.77069 |
| Global Rank | 1,596,775.66 | 1,438,523 | 1,664,022 | 89,032.06 |

N = 180 days for five airline websites and five crowdsourcing websites.

### 3.1. Statistical Analysis

Table 3 illustrates the Pearson's $\rho$ Coefficients for the first hypothesis.

**Table 3.** Coefficients between the metrics for H1.

| | Unique Visitors | Paid Traffic Cost | Organic Traffic Airlines |
|---|---|---|---|
| Unique Visitors | 1 | | |
| Paid Traffic Cost | 0.671 | 1 | |
| Organic Traffic Airlines | 0.445 | 0.885 ** | 1 |

** Correlation is significant at the 0.01 level (1-tailed).

In regard to the first hypothesis (H1), Unique Visitors and Paid Traffic Cost resulted in a positive correlation with $\rho = 0.671$, which means that the paid traffic benefits the websites with more unique visitors. Nevertheless, this correlation is not significant.

Moreover, the Organic Traffic Airlines metric and Paid Traffic Cost metric resulted in a significant positive correlation with $\rho = 0.885 **$, meaning that as the number of Paid Traffic increases, the organic traffic of the airline companies will also experience an increase.

Table 4 demonstrates the Pearson's $\rho$ Coefficients for the second hypothesis.

**Table 4.** Coefficients between the metrics for H2.

| | Organic Traffic Crowdsourcing | Paid Keywords | Unique Visitors |
|---|---|---|---|
| Organic Traffic Crowdsourcing | 1 | | |
| Paid Keywords | 0.907 ** | 1 | |
| Unique Visitors | 0.705 | 0.670 | 1 |

** Correlation is significant at the 0.01 level (1-tailed).

For the second hypothesis (H2), Unique Visitors and Paid Keywords resulted in a positive correlation with $\rho = 0.705$. This correlation signifies that Paid Keywords influence, in a beneficial manner, the websites with more unique visitors. However, this correlation is not significant. Also, the Organic Traffic of crowdsourcing websites metric and Paid Keywords metric resulted in a significant positive correlation with $\rho = 0.907**$, which means that as the number of Paid Keywords increases, the Organic Traffic of crowdsourcing websites increases as well.

Table 5 presents the Pearson's $\rho$ Coefficients for the third hypothesis.

**Table 5.** Coefficients between the metrics for H3.

|  | Organic Traffic Crowdsourcing | Organic Traffic Airlines | Average Visits Duration |
|---|---|---|---|
| Organic Traffic Crowdsourcing | 1 |  |  |
| Organic Traffic Airlines | 0.688 | 1 |  |
| Average Visits Duration | 0.333 | −0.071 | 1 |

Correlation is significant at the 0.01 level (1-tailed).

As for the third hypothesis (H3), Organic Traffic of crowdsourcing websites and Organic Traffic Airlines websites resulted in a positive correlation with $\rho = 0.688$, which means that when the organic traffic from the crowdsourcing websites increases, the organic traffic of the airline companies will also increase. Nevertheless, this correlation is not significant. Furthermore, the Organic Traffic of crowdsourcing websites metric and Average Visits Duration metric resulted in a positive correlation with $\rho = 0.333$, meaning that as the number of the Organic Traffic of crowdsourcing websites increases, the Average Visits Duration spending on airline websites increasing as well.

Interestingly, a negative correlation was identified between the Average Visits Duration metric and the Organic Traffic Airlines metric with $\rho = -0.071$. This means that when the Organic Traffic of the airline's websites increases, the Average Visits Duration spending on airline companies' websites decreases. This signifies that hypothesis H3 is disproven, as the correlation between the examined metrics in question is not significant. The absence of a statistical correlation is an outcome that, on a commonsense basis, was not anticipated.

Table 6 illustrates the Pearson's $\rho$ Coefficients for the fourth hypothesis.

**Table 6.** Coefficients between the metrics for H4.

|  | Organic Traffic Airlines | Average Visits Duration | Global Rank |
|---|---|---|---|
| Organic Traffic Airlines | 1 |  |  |
| Average Visits Duration | −0.071 | 1 |  |
| Global Rank | −0.071 | −0.631 | 1 |

Correlation is significant at the 0.01 level (1-tailed).

For the fourth hypothesis (H4), Global Rank from the airline companies and the Organic Traffic from airline websites resulted in a negative correlation with $\rho = -0.071$, which means that when the Organic Traffic of the airline companies increases, the Global Rank from the airline websites shall also increase. This is not, however, a significant correlation. In addition, the AVD metric and Global Rank metric of the airline companies resulted in a negative correlation with $\rho = -0.631$, meaning that when the Organic Traffic of the airline companies increases, the Global Rank metric of the airline decreases.

The results of the fifth hypothesis (H5) are illustrated in the next section of this study, as the outcome of this hypothesis is a product of the ABM and the User Engagement metric (AVD, Unique Visitors and Pages Per Visits).

Table 7 illustrates the Pearson's $\rho$ Coefficients for the sixth hypothesis, H6.

**Table 7.** Coefficients between the metrics for H6.

|  | Organic Traffic Airlines | Paid Keywords | Paid Traffic Cost |
|---|:---:|:---:|:---:|
| Organic Traffic Airlines | 1 |  |  |
| Paid Keywords | 0.878 * | 1 |  |
| Paid Traffic Cost | 0.885 ** | 0.989 ** | 1 |

** Correlation is significant at the 0.01 level (1-tailed). * Correlation is significant at the 0.02 level (2-tailed).

For the sixth hypothesis (H6), Paid Keywords from the airline companies and the Organic Traffic from airline websites resulted in a positive correlation with $\rho = 0.878$, which means that when the Organic Traffic of the airline companies increases, the Global Rank from the airline websites shall also increase. This is a significant correlation. In addition, the Paid Traffic Cost metric and Organic Traffic Airlines metric of the airline companies resulted in a positive correlation with $\rho = 0.885$, meaning that when the Organic Traffic of the airline companies increases, the Global Rank metric of the airline increases.

### 3.2. FCM Adaptation

In Figure 1 below, the FCM with the statistical analysis intercorrelations is illustrated. Figure 1 was created using the statistical correlations identified in Section 3.1. Blue arrows illustrate positive correlations, while red arrows represent a negative one. The plus sign represents a positive correlation between the two metrics, while the minus sign represents a negative correlation. This methodology was implemented in this paper as it has commonly been used in previous studies regarding Search Engine Optimization (SEO) [59]. For example, as shown in Figure 1, the Unique Visitors of the airline companies can be generated either from Organic Traffic or from Paid Traffic.

### 3.3. Agent-Based Model

The ABM illustrated in Figure 2 contains components constituted by specific characteristics which interact within a "cause and effect" system to produce an output of correlations [41,54]. The use of ABM was selected for the purposes of this study due to its dual emphasis on both the performance of the system components as well as the system output [55]. In practical terms, the use of ABM allows for the observation of both the performance data generated from organic visitor traffic and user engagement with the digital platform [60]. In turn, this creates a data output of the correlations, which result from these different user interactions. The emphasis placed by ABM on the existence and identification of correlations between different system components means that this developmental approach prioritizes the identification of settings and circumstances which influence the user's behavior within the model [61]. Thus, the use of ABM allows companies to fully comprehend the insights provided from the data sets in regard to user engagement with their digital platforms as well as areas of growth [55]. The total model creation relies on the programming language "Java".

Figure 2 illustrates the creation of the ABM through the use of the AnyLogic 8.7.3 simulation system. The ABM provides a visual representation of the influence of display ads on branded search traffic visitors and their subsequent interaction with the examined websites. The oval blocks illustrate the original situation of the agent prior to any transfer occurring. Transitions between blocks are represented by the arrows shown on the model, while such transitions result from alterations to prescribed conditions dictated on the right side of the figure. The prescribed conditions used in the ABM were obtained using Poisson probability from the output of the data analysis previously described. Poisson probability function was selected for the purposes of this research [62] for two reasons. First, it allows for the accurate estimation of the time interval needed to execute the predictive model within the observation period (i.e., 220 continuous days). Second, data from the statistical correlations presented in Section 3.1 are implemented in the model in order for the agents to move from one level to the next.

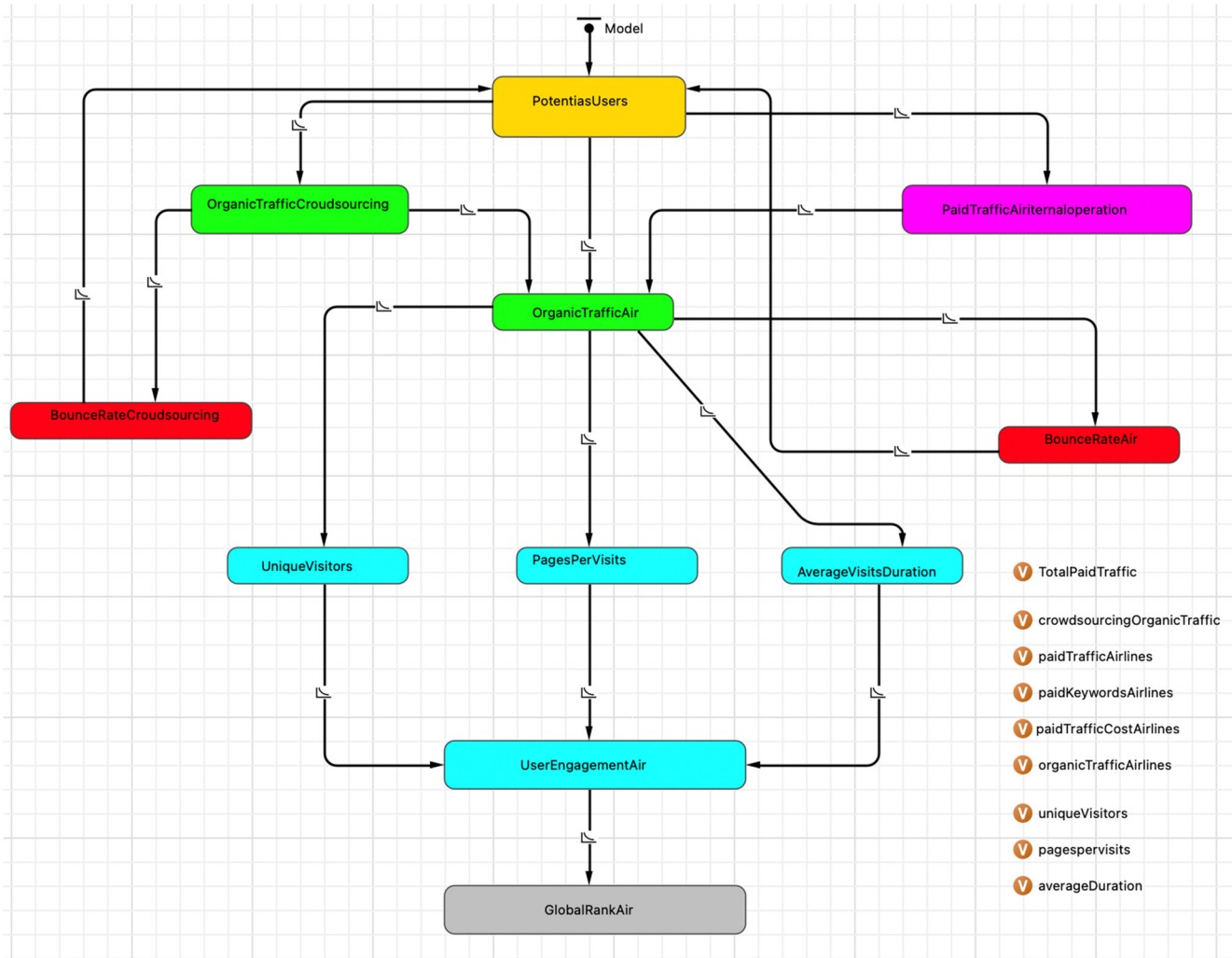

**Figure 2.** Agent-based model development for the simulation and optimization of airline companies' User Engagement and Global Rank through Search Engine Optimization by concentrating on website Organic Traffic.

The model first illustrates, in the top middle box, potential website users who are divided into groups of users who directly access the crowdsourcing website and groups of users who access the airline company website through paid traffic. The paid traffic includes variables depicted on the right-hand side of the model, namely, TotalPaidTraffic, paidTrafficAirlines, paidKeywordsAirlines, paidTrafficCostAirlines. Some users who directly accessed the crowdsourcing websites exited the webpage and were redirected to the original potential website user box by passing through the bounce rate, while other users access the OrganicTrafficAir, wherein some will exit the page through the bounce rate and others will provide data for the User Engagement parameters (Unique Visitors, Average Visits Duration, Pages per Visits). The insights provided by the User Engagement parameters (UserEngagementAir) provides an estimation of the Global Rank of the airline companies examined. The output of the ABM generates the agent population illustrated in Figure 3a,b below.

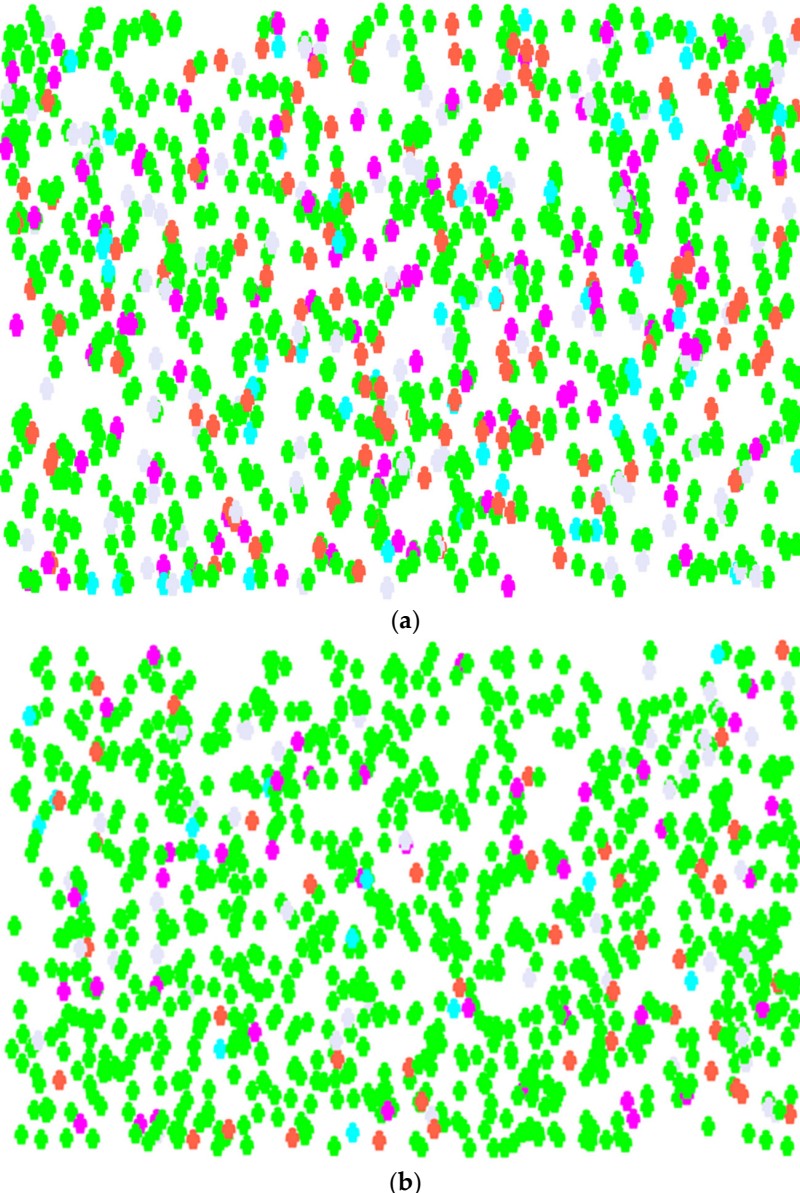

**Figure 3.** (**a**) Population allocation in experiment with 1000 agents for a period of 220 days. Day 40. (**b**) Population allocation in experiment with 1000 agents for a period of 220 days. Day 150.

Figure 3a,b illustrates the operation of the ABM model represented in Figure 2. The study investigates the flow of the users in the model for an interval of 220 days. The agents represent the potential users running and operating throughout the process until they reach the final box coined "GlobalRankAir" in Figure 2. The green agents represent the Organic traffic users, the purple agents show users generated from Paid Traffic, the blue agents represent the visitors of the website that contribute to User Engagement and the gray agents represent the visitors of the website that contribute to the growth of the website Global Rank. Finally, the red agents represent the bounce rate amount of agent. In Figure 3a, day 40 of 220 is depicted and shows that there are approximately equal agents from organic traffic (green) and Paid Traffic (purple). Figure 3b was obtained from day 150 of 220 and shows that when the model is running, there are more agents from organic traffic (green) than from Paid Traffic (purple). This is expected, because after the paid traffic campaign has been initiated, the company's Global Rank has been raised, and the users reach the company's website directly, given that the corporate brand has grown.

On the vertical axis, the time charts (Figures 4–11) indicate the value of the results, either User Engagement, Organic Traffic, Traffic Cost or Global Rank, while the horizontal axis represents the time range during which the simulation occurs (i.e., 220 continuous days) for the five crowdsourcing websites and five airline websites. The time charts *"display the history of contribution of an amount of data into a total during the latest time horizon as stacked areas"* [63].

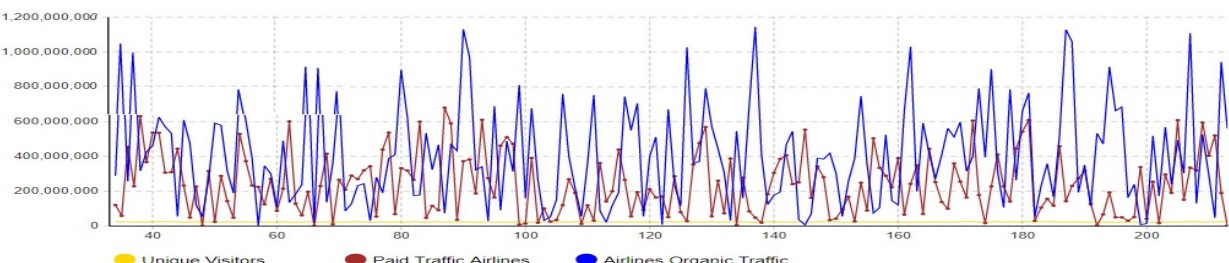

**Figure 4.** Time chart which illustrates the history of contribution of airline companies' Organic Traffic and Unique Visitors with Paid Traffic. Vertical axis indicates the value of the results. Horizontal axis represents the time range during which the simulation occurs.

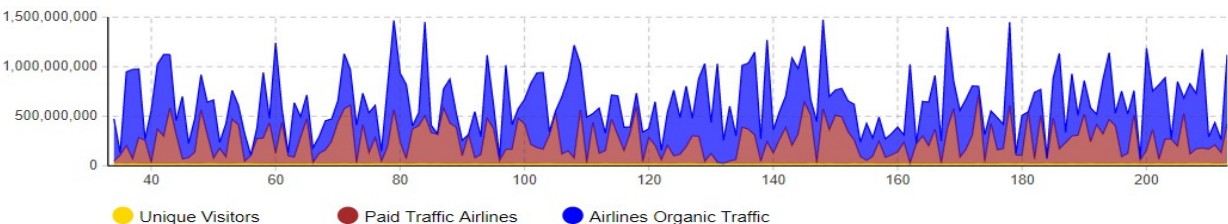

**Figure 5.** Time chart which illustrates the history of contribution of airline companies' Organic Traffic and Unique Visitors with Paid Traffic. Vertical axis indicates the value of the results. Horizontal axis represents the time range during which the simulation occurs.

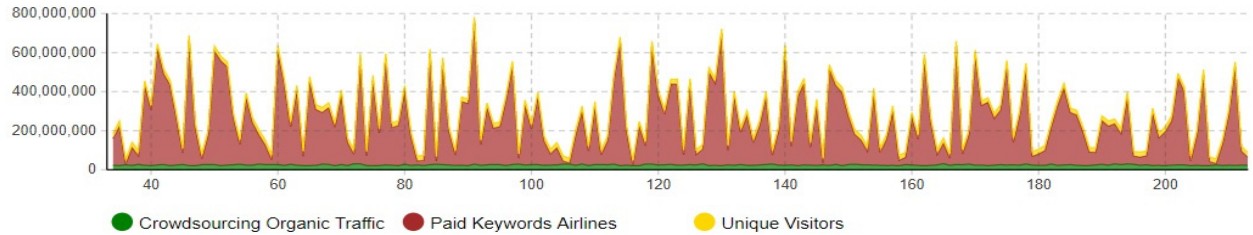

**Figure 6.** Time chart which illustrates the history of contribution of airline companies' Paid Keywords and Unique Visitors with Crowdsourcing Organic Traffic. Vertical axis indicates the value of the results. Horizontal axis represents the time range during which the simulation occurs.

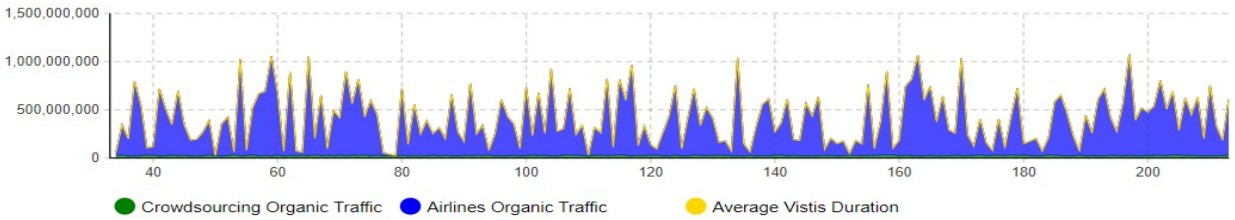

**Figure 7.** Time chart which illustrates the history of contribution of airline companies' of airline companies' Organic Traffic and Average Visits Duration with Crowdsourcing Organic Traffic. Vertical axis indicates the value of the results. Horizontal axis represents the time range during which the simulation occurs.

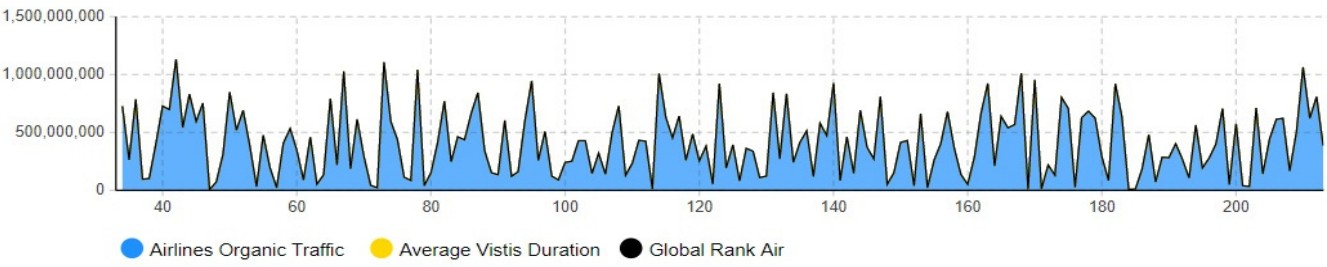

**Figure 8.** Time chart which illustrates the history of contribution of airline companies' Average Visit Duration and Organic Traffic with Global Rank of airline companies. Vertical axis indicates the value of the results. Horizontal axis represents the time range during which the simulation occurs.

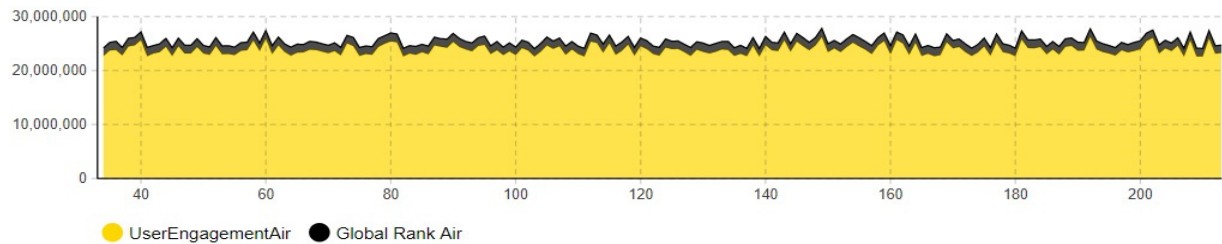

**Figure 9.** Time chart which illustrates the history of contribution of airline companies' User Engagement, as extracted from the model, with Global Rank of airline companies. Vertical axis indicates the value of the results. Horizontal axis represents the time range during which the simulation occurs.

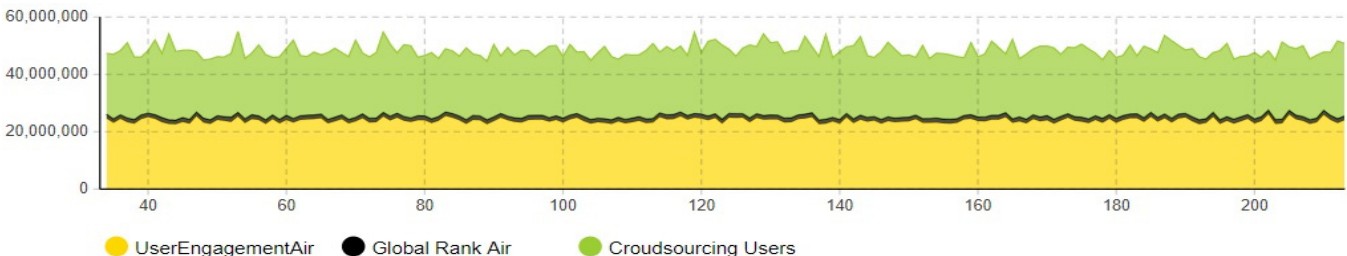

**Figure 10.** Time chart which illustrates the history of contribution of airline companies' User Engagement and Global Rank with the total Crowdsourcing Users. Vertical axis indicates the value of the results. Horizontal axis represents the time range during which the simulation occurs.

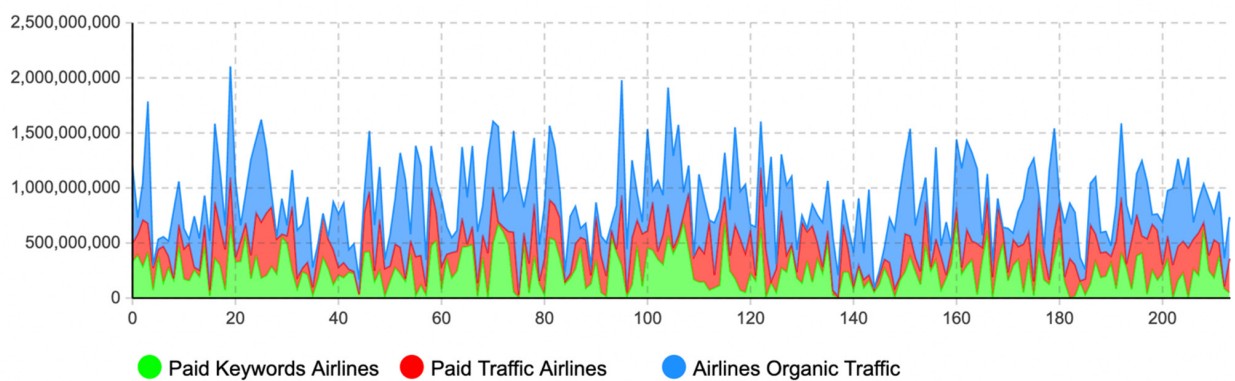

**Figure 11.** Time chart which illustrates the history of contribution of airline companies' Organic Traffic with the Paid Keywords and the Paid Traffic. Vertical axis indicates the value of the results. Horizontal axis represents the time range during which the simulation occurs.

For the first hypothesis (H1), the following findings were observed in Figures 4 and 5. After running the ABM, the research findings seem to be identical with the statistical results:

(1) It was observed that after a paid traffic campaign was initiated, a peak occurred in the organic traffic. This reveals a pattern wherein, following a specific paid campaign, a boost in Organic Traffic occurs. After the 62-day mark, the companies create a Brand Name, which results in the companies experiencing high organic traffic results without having to invest as much in paid campaigns. This phenomenon serves to further verify the existence of an established Brand Name.

(2) The time charts further reveal that when the Paid Traffic remains constant, the peaks of the organic traffic variables become considerably greater, indicating that a Brand Name has been established.

As this is a time stack chart created by Anylogic [63], it illustrates the contribution of the examined metrics. Figures 4 and 5 show that the Unique Visitors metric has a low contribution in regard to the Paid Traffic Airlines metric (which is why Figures 4 and 5 show a low constant yellow line on the bottom of the time stack chart). Contrastingly, the Paid Traffic Airlines metric has a high contribution to the Airlines Organic Traffic metric, as depicted in Figures 4 and 5. It can further be observed that following each paid traffic campaign (illustrated in Figures 4 and 5 as a spike), there is an amplified peak which is mirrored on the Airline's Organic Traffic.

In relation to hypothesis (H2), after running the ABM, the research findings seem to be identical to the statistical results, as illustrated in Figure 6. The statistical analysis reveals a positive correlation between the Paid Keywords and the Unique Visitors variables for the airline companies. Indeed, the highest recorded values on the chart occurred on days 117, 129 and 191. In respect of "Crowdsourcing Organic Traffic", the parameter does not remain constant throughout the observation period, but instead is much lower than the remaining two parameters (i.e., Paid Keywords Airlines and Unique Visitors) and, as such, does not appear on the time chart shown in Figure 6. Moreover, since the time chart illustrates the history of contribution between the examined metrics, it can be inferred that the unique visitors and the paid keywords variables have a high degree of correlation between them. Figure 6 illustrates that the "Crowdsourcing Organic Traffic" metric has a low contribution to the Paid Keywords Airlines metric. Contrastingly, the Paid Keywords Airlines metric has a high contribution on the Unique Visitors metric as the peaks in the Paid Keywords airline metrics are mirrored in the Unique Visitors metric (Figure 6). This means that following a paid campaign, more unique visitors enter the airline company's website.

Figure 6 displays very small fluctuations in the number of Unique Visitors. This signifies that both Paid Keywords Airlines and Crowdsourcing Organic Traffic result in repetitive visitors, contributing to the establishment of the corporate brand name. The ABM suggests that by relying on this digital marketing strategy, companies do not generate many new visitors. Nevertheless, small increases in new visitors remain commercially important and desirable, which is why they are being examined.

In respect of the third hypothesis (H3), after running the ABM, the research findings seem to be different with the statistical results, as illustrated by Figure 7 below. As previously identified from the results of the statistical analysis, the simulation model reveals a positive correlation between airline companies' Organic Traffic and Average Visits Duration. The highest values on the charts occurred on days 58, 162 and 191.

Regarding the fourth hypothesis (H4), the research findings seem to differ from the statistical results, as illustrated by the time charts below (Figures 8 and 9). The following findings were observed:

(1) The simulation model illustrates a positive correlation between Global Rank and the Organic Traffic of the airline companies. The highest values on the chart occurred on days 41, 72 and 217.

(2) While the AVD metric does not have a positive correlation with the Global Rank metric, all the User Engagement metrics (Pages per Visits, Average Visit Duration and Unique Visitors) have a positive correlation with the Global Rank. This finding is further

reinforced by the time chart shown in Figure 9, which illustrates a considerable correlation in the fluctuations which occur between the "User Engagement" and "Global Rank" metrics. This suggests that perhaps the Google algorithm also relies on the correlations between the User Engagement variables shown in Figure 9. This, however, is only one of many possibilities. The secrecy of the Google algorithm used to determine the rankings of the Global Rank does not allow us to make an accurate deduction. Nevertheless, the ABM authenticates findings of previous research which maintain that User Engagement contributed to the increase in rank of the corporate website's Global Rank [38,41,59].

In Figure 8, the AVD contribution is much lower than the remaining two variables (Airlines' Organic Traffic and Global Rank Air), which is why it cannot be visualized, since the time chart illustrates the history of contribution on a time scale [63].

Finally, for hypothesis (H5), Figure 10 shows a clear correlation between the Global Rank of airline companies and the User Engagement of the airline companies. Nevertheless, no correlation was found to exist between these two variables (User Engagement and Global Rank) and the total crowdsourcing users metric [43]. This shows that either the ads of the airline companies on the crowdsourcing websites are not placed correctly, or that the tasks assigned to the crowdsourcing websites, through the airline companies, are not appealing enough to attract the user's attention.

Finally, for the sixth hypothesis (H6), after running the ABM, the research findings appear to be identical with the statistical results. Figure 11 shows a clear correlation between the Paid Traffic and Paid Keywords of the airline companies with the Organic Traffic. Figure 11 shows that when airline companies invest in Paid Traffic, their Organic Traffic increases. The implicit participation process consists of the users of the crowdsourcing or crowdfunding websites seeing an advertisement of an airline company on the crowdsourcing or crowdfunding website, selecting the advertisement and being redirected to the airline company's website.

## 4. Discussion

This study's goal is to create an innovative methodological approach in order to examine the impact of Organic Traffic, User Engagement and Global Rank in order to consider approaches which can be implemented to improve digital marketing strategies. In addition, this paper sought to create a computational-data-driven, three-stage methodology in order to estimate the user engagement and brand name impact between five crowdsourcing and five airline companies' websites.

It was subsequently shown that branded search traffic can be utilized as a means of enhancing user engagement within a given digital platform through an increase in the Pages per Visits, Average Visit Duration and Unique Visitors variables. The study further identified the existence of positive correlations between the User Engagement metrics and the Global Rank. These findings were further supported by the results of the data analysis which reiterated the existence of the identified correlations between the examined metrics. The identification of these variables was a significant insight, as it provided a thorough understanding of the Big Data analytics and permitted the subsequent use of these insights to improve existing digital marketing strategies [64,65]. In essence, the methodology adopted within this study achieved three goals: (a) the thorough understanding of the data analytics metrics and the outputs of the data analysis of said metrics, (b) the identification of existing correlations between the examined metrics and (c) the use of these insights to best comprehend user behavior within the scope of digital marketing strategies.

## 5. Conclusions

This study gathered and analyzed airline website data, which led to the discovery of interesting insights, which can be used by marketeers, developers and decision makers to improve digital marketing strategies. The study focused on User Engagement and Organic Traffic.

### 5.1. Data-Driven ABM Development and Contribution to Web Analytics

ABM is a tool which was used to examine and predict the specific time intervals during the examined period of 220 continuous days within which there was the highest rate of search traffic generated from time spent and pages per views. It should be noted that these correlations are not static as they originate from behavioral analysis, which is intrinsically dynamic [38,41,64,66]. This characteristic renders ABM suitable for: (a) organizations exploiting the Big Data insights relating to user engagement with digital marketing strategies and crowdsourcing websites, (b) emphasizing user behavior in web analytics platforms in order to ensure the integration of these characteristics during the ABM development [51] and (c) understanding the suitability of advertising strategies.

The transformation of the behavioral analytics into tangible digital marketing strategies is a difficult task that includes a lot of different parameters, namely, establishing Key Point Indicators (KPIs) [67,68]. This study presented some interesting findings regarding the interactions between crowdsourcing websites and airline companies' websites. In respect of the first and the second hypotheses (H1, H2), the outcomes of this research verified the creation of a brand name following the deployment of paid campaigns from the airline companies to crowdsourcing websites. The deployment of these campaigns was also found to have a positive impact on user engagement. This finding aligns with existing research conducted in respect of other sectors and industries [38,41]. Hypotheses H3 and H4 revealed a positive correlation between User Engagement and Organic Traffic, as well as a positive correlation between User Engagement and Global Rank. Once more, these finding fully aligned with existing research on the topic, albeit previous research was conducted in respect of different sectors than those examined in this paper [59]. These findings corroborate previous literature and may suggest that the Google algorithm takes into consideration the totality of the User Engagement metrics and not its individual components (i.e., Average Visit Duration, Pages per Visit) [38,41,59]. Hypothesis (H5) verifies that there was no significant correlation between the total users of crowdsourcing as gathered from the Deloitte research [43] and the Global Rank or User Engagement of the airline companies' websites.

### 5.2. Implicit Participation and Web Analytics

Another objective of this study was to illustrate the implicit participation procedure of potential website users in the hypothesis (H6). The findings aligned with existing research in the field [37,41]. Both statistical analysis and the ABM presented a strong correlation between the Paid Traffic and Paid Keywords with the Organic Traffic of the airline companies. Moreover, much like previous research, this study can conclude that user engagement and participation is influenced by paid campaigns [38,41].

### 5.3. Future Research

The focus of this study was to examine, consider and evaluate the contribution of crowdsourcing platforms' traffic to airline companies' user engagement and its later effect on their website traffic and brand name. This objective was attained through the use of efficient predictive model and the use of available mass-mode website Big Data. A statistical analysis was carried out in order to illustrate the correlations between the examined metrics. The methodological process used within this study can be further used in the green logistics sector in order to verify the good standing of existing research results obtained through the use of a different methodological process, in order to examine the reliability and efficiency of existing methodological processes in logistics research [69,70]. Indeed, due to the innovative nature of the adopted methodological framework, further research should be conducted relying on such an approach in order to verify the credibility of this type of methodological process. While this is certainly an exciting area for future investigation due to its existing validating need, the adopted framework and its subsequent output provided reliable results within the scope of this study.

A further opportunity for research is the consideration of the ABM's and the Fuzzy Cognitive Maps' versatility and adaptivity. Fuzzy Cognitive Mapping is often relied on as a mean of adopting a macro-level approach to data analysis, while ABM is used to provide a more micro-level perspective to the analysis. The common reliance on these types of models and the overwhelming nature of Big Data and web analytic metrics renders it imperative for ABM and Fuzzy Cognitive Maps to provide an efficient means of measuring the performance of a digital platform. Finally, future research could incorporate neuromarketing analysis in order to gather data of participation and user engagement. Neuromarketing analysis should be conducted in this field to obtain a greater understanding of user emotion and behavior. Techniques such as *Electroencephalogram* (EEG) which is an assessment of brain activity using sensors [71] and Functional Magnetic Resonance Imaging (FMRI), which records even minor changes in blood flow that take place in the brain [71], could provide an interesting avenue for future research in this area. Two crucial elements of participation are obtaining information and giving information [71], and that is the reason why further research with EEG can provide useful insights for the process of user participation and human–machine interaction.

*5.4. Limitations*

While this study provided significant insights into the impact of organic traffic on user engagement and may contribute to the existing research in the field, this study had certain limitations which ought to be considered to carry out further research on this topic.

First, this study relied on a limited number of digital platforms on which the data analysis was conducted, and as such, the results obtained may provide a narrow view of the larger picture. Notwithstanding, the five crowdsourcing and crowdfunding platforms which were selected for this paper are the most important crowdsourcing and crowdfunding platforms as they maintain relationships between crowdsourcing websites and airline company websites. Still, future research on the topic should expand the breadth of examined platforms, not only in respect of the number of platforms selected but also the types of platforms examined.

Second, the data used to carry out our analysis was gathered from a web analytics data gathering website known as "SEMRush". A more holistic approach could have been adopted for this study by relying on more than one web analytics data gathering websites, such as Alexa, in order to obtain a wider pool of data to be examine.

Finally, this study focused on a single industry, the commercial airline industry, when considering the impact of organic traffic on user engagement. Future research should consider extending the scope of examination to include the wider logistics industry in order to obtain a thorough understanding of the impact of organic traffic on the wider industry at large. Conversely, this study limited its scope in the analysis of the airline industry by focusing on the five major commercial airline companies. The examined airline companies were selected as they undoubtedly provide an overview of the wider commercial airline industry. Still, increasing the number of airline companies examined could provide a more precise and accurate depiction of the commercial airline industry, thus acquiring greater insights in this area of research.

**Author Contributions:** Conceptualization, D.P.S. and D.P.R.; methodology, D.P.S. and D.P.R.; software, D.P.R.; validation, D.P.R.; formal analysis, D.P.R.; investigation, D.P.R.; resources, D.P.S. and D.P.R.; data curation, D.P.R.; writing—original draft preparation, D.P.R.; writing—review and editing, D.P.R.; visualization, D.P.R.; supervision, D.P.S.; project administration, D.P.R. All authors have read and agreed to the published version of the manuscript.

**Funding:** This research received no external funding.

**Institutional Review Board Statement:** Not applicable.

**Informed Consent Statement:** Not applicable.

**Data Availability Statement:** Not applicable.

**Conflicts of Interest:** The authors declare no conflict of interest.

## Abbreviations

| | |
|---|---|
| ABM | Agent Based Model |
| AVD | Average Visit Duration |
| EEG | Electroencephalogram |
| FCM | Fuzzy Cognitive Map |
| FMRI | Functional Magnetic Resonance Imaging |
| SEM | Search Engine Marketing |
| SEO | Search Engine Optimization |

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
