# Peer review of "The Impact of Organic Traffic of Crowdsourcing Platforms on Airlines’ Website Traffic and User Engagement"

_sustainability, doi:10.3390/su13168850_

Round 1

Reviewer 1 Report

First, I would like to congratulate the authors on their work. The topic of the paper is significantly important in the crowdsourcing domain.  The paper has a good organization, and the structure of the paper is well defined. The paper presents a good research background, and the research design, hypotheses, and used methods, including the statistical analysis, were clearly stated. The presentation of the research is clear and very readable. The results are interesting and significant. The references are relevant and up-to-date.

Nevertheless, some significant elements are missing. The authors should include the section Conclusion and describe the limitations of the research.

Author Response

Dear Sir/Madam, 

I would firstly like to take this opportunity to sincerely thank you for reviewing my paper and providing me with constructive feedback. Below please find my responses which address your comments. 

  1. "The authors should include the section Conclusion": A conclusion has been added to the paper and can be seen at line starting from 726 and ending at line 851.
  2. "Describe the limitations of the research":The limitations have been described in section 5.5 entitled "Limitations" which can be found at lines 819 to 851. 

Once again, I would like to thank you for your review and for the opportunity to submit research for your journal. By addressing your concerns, I was able to provide the paper with a greater degree of detail. 

Sincerely yours,

The author

Reviewer 2 Report

This paper uses crowdsourced data to make evident the impact of certain variables through five hypotheses. The authors present a predictive model to improve organic traffic and user engagement. However, the results remain somewhat unclear. Further, there is a variety of flaws in research design and execution, as well as a variety of claims that stay unproven.

As a minor point, the subject of this paper is somewhat in the outskirts of the term "crowdsourcing". Usually, crowdsourcing projects have a goal for the benefit of a community or the public. Here, the data are retrieved by implicit piggyback crowdsourcing, i.e., third parties make use of data that the "crowd" provides. From a philosophical point of view, this type of data collection would be in the outskirts of the term crowdsourcing; however, in a wider sense, authors within smart cities and marketing tend to extend the term "crowdsourcing".

Several claims appear to be rather generic and difficult to grasp. For instance. line 134ff., all three points need to be rewritten. It is unclear what you mean by outsourcing projects in this particular case. Further, what would be appropriate crowdsourcing platforms? The second point is imprecise and rather vague, as is the third. What would be the difference between these three?

The term "organic traffic" is a construct that is also explained in Table 1. However, it is not defined which metric to use for this construct. Can it be decomposed as a construct of several variables? How is the engagement construct defined? Line 153 gives some examples, but the construct needs to be defined more precisely; e.g., which formula would be suitable for an indicator?

What is "crowd branding"? What do you mean by "incidentally" in Line 156?

incomprehensible, line 154: benefits afforded to airline companies

H1 and H2 seem to use the same variables, with the impact going the opposite way. Did I understand this correctly? I am not sure if I understood the different impacts of H1 and H2. Further, for H2, it is not evident how the "engagement" construct comes into the picture, as H2 connects organic traffic, paid keywords, and unique visitors.

Line 169: incomprehensible sentence. It is further unclear how the "importance" was verified by using "eye-tracking"; also the differences on the websites are not further taken into account (at least, it is not described).

Also, H4 and H5 are difficult to understand. Line 193: what are the "User Engagement" metric and the other metrics? Please describe shortly what these are about, so that your article becomes self-contained. 

Please add a table of acronyms in the appendix. ABM, FCM (acronym should be introduced earlier), and others.

Line 216: definition "organic traffic": incomprehensible. What are visitors that are generated? Further, there is a language issue with this sentence.

Average Visits Duration: replace "can be defined" --> "is defined". Further, why use the average here? "Visit duration" would be appropriate. The averaging process is dependent on how you assess.

Headline 2.3 sounds strange. Further, why is this a simulation? It should be a model. In your paper, you don't describe a simulation.

Section 2.4 is incomprehensible. Further, eye tracking would only be a minimal part of neuromarketing; thus, eye-tracking would suffice to characterise what you describe in Section 2.4.

line 262: remove "hereinabove"

Table 2 is incomprehensible, both caption and contents. For some of the values, it is unclear how it is measured. Further, when calculating mean, min, max: what is it the mean value (etc) of? Is this a mean value per day, or per company, or something else? What is the difference between organic traffic crowd(sourcing) and organic traffic airlines?

Tables 3 and following are symmetric. What does this mean? Is this an impact with a direction or simply a correlation? But then, Tables 3 and 4 should be identical, as just the direction is exchanged?

Line 284: Table5 present --> Table 5 presents

Line 288ff: are these values significant? It is further unclear what these findings mean.

Line 312: Do you mean "Adaptation"?

Which process was used to create Figure 1? (you only mention the tool, but not the process).

Section 3.2 is rather generic. Further, lines 338-342: incomprehensible. What do you mean by "descripted statistics"?

The paragraph starting with Line 347: incomprehensible. How can a model "begin"? user --> users. Suggestion "divided into user" --> "divided into groups of users"

Line 351: "left the page though the bounce rate" does not make sense.

Figure 3: Looks like art, but there is no explanation of what the colours mean and what is on the axes, and which units to use.

Line 360: which statecharts?

Line 364ff: incomprehensible. What does "the above statistical analysis is confirmed" mean?

Line 365: Figures 4 and 5

Line 367: "was" missing: campaign was initiated

Figure 4 and following: These figures are not distributions, but some type of time series. Why is the number of unique users constant? I don't understand this. Which units are used on the axes? is the x-axis the day number?

Figure 6: why is the crowdsourcing organic traffic constant? Why is the unique visitors identical with the paid keywords airlines?

Line 403: unclear how you can draw this conclusion. There are other possibilities.

Figure 8: average visits duration is missing.

Figure 9: incomprehensible. Why are these values equal?

Section 3.3 is inappropriately described. How were the candidates recruited? How was the experiment performed? Which data was collected? What were the criteria and metrics to be used? It seems that you used some tool, but it Is unclear which mechanisms were used. How does Figure 10 show what you wanted to show? A user gazing at an area in-between information elements does not prove anything.

Line 461: Does ABM really predict the optimal number of display ads? It is not described as such in the paper.

Line 486: As you don't know how the Google algorithm is constructed, you cannot make this assumption from the data you collected.

Line 522ff: Performing EEG and similar methods are interesting, but not within the scope of your paper. Please describe how these technologies would be an extension of your research. Please note, that it is not evident yet how you used eye-tracking and what it proves in your particular case.

Author Response

Dear Sir/Madam, 

Thank you for your review of my paper and for providing me with constructive feedback. Below please find a point-by-point response to your comments.

  1. "As a minor point, the subject of this paper is somewhat in the outskirts of the term "crowdsourcing": This comment has been addressed at lines 50-53.
  2. "From a philosophical point of view, this type of data collection would be in the outskirts of the term crowdsourcing": This comment has been addressed at lines 120-123. 
  3. "Line 134ff., all three points need to be rewritten. It is unclear what you mean by outsourcing projects in this particular case. Further, what would be appropriate crowdsourcing platforms? The second point is imprecise and rather vague, as is the third. What would be the difference between these three?": Each point has been addressed in lines 196 to 217.
  4.  
    1. "The term "organic traffic" is a construct that is also explained in Table 1. However, it is not defined which metric to use for this construct. Can it be decomposed as a construct of several variables": This has been defined in Table 1 under the metric coined "Organic Traffic" which can be found at line 337. Furthermore, organic traffic is a single metric generated directly from SEMRush, as stated in Table 1.  
    2. How is the engagement construct defined? Line 153 gives some examples, but the construct needs to be defined more precisely; e.g., which formula would be suitable for an indicator?": See lines 179 to 181.
  5. "What is "crowd branding"? What do you mean by "incidentally" in Line 156?": See lines 230-233.
  6. "Incomprehensible, line 154: benefits afforded to airline companies": Addressed at lines 228-230.
  7. "H1 and H2 seem to use the same variables, with the impact going the opposite way. Did I understand this correctly? I am not sure if I understood the different impacts of H1 and H2. Further, for H2, it is not evident how the "engagement" construct comes into the picture, as H2 connects organic traffic, paid keywords, and unique visitors": H1 and H2 have been rewritten so as to make the difference between the two hypotheses evident. This can found at lines 222 to 239. 
  8. "Line 169: incomprehensible sentence. It is further unclear how the "importance" was verified by using "eye-tracking"; also the differences on the websites are not further taken into account (at least, it is not described)": A new hypothesis (H7) has been formulated and can be seen at line 277-293.  
  9. "Also, H4 and H5 are difficult to understand. Line 193: what are the "User Engagement" metric and the other metrics? Please describe shortly what these are about, so that your article becomes self-contained": H4 and H5 have been amended so as to make them more clear and can be found at lines 246-260. As for the user engagement metric please refer to lines 179 to 181.
  10. "Please add a table of acronyms in the appendix. ABM, FCM (acronym should be introduced earlier), and others": A table of acronyms has been added to the Appendix which can be found at line 845.
  11. "Line 216: definition "organic traffic": incomprehensible. What are visitors that are generated? Further, there is a language issue with this sentence": Addressed at lines 178-181.
  12. "Average Visits Duration: replace "can be defined" --> "is defined". Further, why use the average here? "Visit duration" would be appropriate. The averaging process is dependent on how you assess":  An explanation has been provided in Table 1 under the metric "Average Visit Duration" and can be found at lines 337-338.
  13. "Headline 2.3 sounds strange. Further, why is this a simulation? It should be a model. In your paper, you don't describe a simulation": Addressed. See lines 351-358.
  14. "Section 2.4 is incomprehensible. Further, eye tracking would only be a minimal part of neuromarketing; thus, eye-tracking would suffice to characterise what you describe in Section 2.4": See lines 376-384.
  15. "Line 262: remove "hereinabove": Done. 
  16. "Table 2 is incomprehensible, both caption and contents. For some of the values, it is unclear how it is measured. Further, when calculating mean, min, max: what is it the mean value (etc) of? Is this a mean value per day, or per company, or something else? What is the difference between organic traffic crowd(sourcing) and organic traffic airlines?": See lines 399-411.
  17. "Tables 3 and following are symmetric. What does this mean? Is this an impact with a direction or simply a correlation? But then, Tables 3 and 4 should be identical, as just the direction is exchanged?": It has been amended, it is a correlation. See lines 412-475. 
  18. "Line 284: Table5 present --> Table 5 presents": Done see line 433. 
  19. "Line 288ff: are these values significant? It is further unclear what these findings mean": See lines 439-450.
  20. "Line 312: Do you mean "Adaptation"?": Yes, changed at lines 476.
  21. "Which process was used to create Figure 1? (you only mention the tool, but not the process)": Described at lines 477-488.
  22. "Section 3.2 is rather generic. Further, lines 338-342: incomprehensible. What do you mean by "descripted statistics"?": Lines 511-516. Also, descripted statistics was changed to statistical correlations.
  23. "The paragraph starting with Line 347: incomprehensible. How can a model "begin"? user --> users. Suggestion "divided into user" --> "divided into groups of users": Done, lines 520-522.
  24. "Line 351: "left the page though the bounce rate" does not make sense": Amended at lines 525-526.
  25. "Figure 3: Looks like art, but there is no explanation of what the colours mean and what is on the axes, and which units to use": Explanation provided at lines 533 to 546.
  26. "Line 360: which statecharts?": See lines 551 to 556.
  27. "Line 364ff: incomprehensible. What does "the above statistical analysis is confirmed" mean?": Addressed at lines 558-559 (It means that it is identical with the statistical analysis).
  28. "Line 365: Figures 4 and 5": Added at line 558. 
  29. "Line 367: "was" missing: campaign was initiated": Added at line 561.
    1. "Figure 4 and following: These figures are not distributions, but some type of time series. Why is the number of unique users constant? I don't understand this": See lines 596-601. 
    2. "Which units are used on the axes? is the x-axis the day number?": Lines 551-556. 
  30. "Figure 6: why is the crowdsourcing organic traffic constant? Why is the unique visitors identical with the paid keywords airlines?": 583-589.
  31. "Line 403: unclear how you can draw this conclusion. There are other possibilities": See lines 623-628.
  32. "Figure 8: average visits duration is missing": Added see lines 629-631.
  33. "Figure 9: incomprehensible. Why are these values equal?": Explained at lines 618-625 and further lines 554 to 556. 
  34. "Section 3.3 is inappropriately described. How were the candidates recruited? How was the experiment performed? Which data was collected? What were the criteria and metrics to be used? It seems that you used some tool, but it Is unclear which mechanisms were used. How does Figure 10 show what you wanted to show? A user gazing at an area in-between information elements does not prove anything.": Section 3.4 has been rewritten to address this comment. See lines 666 to 692.
  35. "Line 461: Does ABM really predict the optimal number of display ads? It is not described as such in the paper": Removed this was added by accident. 
  36. "Line 486: As you don't know how the Google algorithm is constructed, you cannot make this assumption from the data you collected": See lines 745 to 748.
  37. "Line 522ff: Performing EEG and similar methods are interesting, but not within the scope of your paper. Please describe how these technologies would be an extension of your research. Please note, that it is not evident yet how you used eye-tracking and what it proves in your particular case": See lines 793 to 803. 

I trust that the above addresses your comments. I thank you, once more, for reviewing my research. 

Sincerely yours, 

The author

Reviewer 3 Report

Thank you for the opportunity to review the paper "The impact of organic traffic of Crowdsourcing platforms on Airlines’ website traffic and user engagement". The paper addresses an interesting and novel theme. However, there are some observations that should be addressed in this revision.

  1. The theoretical part is incomplete because when it comes to crowdsourcing, there are two important aspects that need to be taken into consideration: participatory culture and mind-sharing, two concepts that are not addressed in this article.
  2. The theoretical background should include them because crowdsourcing happens through participation and the need to participate. At first, an individual doesn’t need to be part of a specific crowd and through participatory culture, the individual becomes part of the crowd.
  3. The link between big data and crowdsourcing is vague since the link between those two is also defined by participatory culture and the way that individuals participate in creating the cultural content, big data is defined by implicit and explicit participation, and that should be one of the theoretical key points of this article.
  4. Since the theoretical background is incomplete, the methodological part is incomplete as well and there are some gaps in the chosen fields because the research focuses on engagement and not at all on participation, crowdsourcing is about engagement and participation as well and how people participate in different situations and how they interact with posts and sites, but engagement is not enough, they have to participate, too.
  5. Regarding the sample, one of the chosen platforms is actually a crowdfunding platform (Indiegogo), not a crowdsourcing platform, which means that there are no crowdsourcing situations on that platform. The rest of the chosen platforms cannot be named crowdsourcing platforms because they are not used only for crowdsourcing, they have other purposes too, not every interaction on those platforms is a crowdsourcing situation.
  6. The chosen hypotheses should rely on participation too, engagement and participation are two different concepts and crowdsourcing is based on user participation as well, the process is not possible without participation.
  7. A participation hypothesis would have made the study complete, in that way there would have been a chance to measure the process of participation amongst the users taking in the account the proposed metrics.
  8. The chosen metrics cover a part of the process but in order to have an exhaustiveanalysisthe used platform should have been taken into consideration when it comes to interface and the degree of use. Usually, the used platform has a big impact on engagement and participation, the research could have measured how the platform impacts the engagement, this could have been an important metric, and it would’ve complete the research and it would’ve opened a new comparison topic and in that way the researched impact would have been more detailed depending on the platform.
  9. The crowdsourcing platform is an important part of this study, the References used could have contained more than one source about the crowdsourcing platforms and the subject of the platforms could have been addressed as a theoretical background, the article could have been covered what a crowdsourcing platform is and how it is defined in this article. Please see Mindsharing: The Art of Crowdsourcing Everything by Zoref, Lior; Gad Allon, Volodymyr Babich (2020) Crowdsourcing and Crowdfunding in the Manufacturing and Services Sectors. Manufacturing & Service Operations Management 22(1):102-112. https://doi.org/10.1287/msom.2019.0825
  10. The whole crowdsourcing process is based on mind-sharing, besides the fact that the article should have taken into consideration that concept, the References would have been more complete with at least a mind-sharing source.

Author Response

Dear Sir/Madam, 

I would firstly like to take this opportunity to thank you for taking the time to review my research and providing me with feedback. Below please find a point-by-point response to your comments. 

  1.  "The theoretical part is incomplete because when it comes to crowdsourcing, there are two important aspects that need to be taken into consideration: participatory culture and mind-sharing, two concepts that are not addressed in this article": See lines 56 to 82.
  2. "The theoretical background should include them because crowdsourcing happens through participation and the need to participate. At first, an individual doesn’t need to be part of a specific crowd and through participatory culture, the individual becomes part of the crowd": See lines 56 to 82. 
  3. "The link between big data and crowdsourcing is vague since the link between those two is also defined by participatory culture and the way that individuals participate in creating the cultural content, big data is defined by implicit and explicit participation, and that should be one of the theoretical key points of this article": Lines 159 to 174. 
  4. "Since the theoretical background is incomplete, the methodological part is incomplete as well and there are some gaps in the chosen fields because the research focuses on engagement and not at all on participation, crowdsourcing is about engagement and participation as well and how people participate in different situations and how they interact with posts and sites, but engagement is not enough, they have to participate, too": Lines 169 to 174. 
  5. "Regarding the sample, one of the chosen platforms is actually a crowdfunding platform (Indiegogo), not a crowdsourcing platform, which means that there are no crowdsourcing situations on that platform. The rest of the chosen platforms cannot be named crowdsourcing platforms because they are not used only for crowdsourcing, they have other purposes too, not every interaction on those platforms is a crowdsourcing situation": Lines 320 to 330. 
  6. "The chosen hypotheses should rely on participation too, engagement and participation are two different concepts and crowdsourcing is based on user participation as well, the process is not possible without participation": Lines 272 to 275. 
  7. "A participation hypothesis would have made the study complete, in that way there would have been a chance to measure the process of participation amongst the users taking in the account the proposed metrics": A new hypothesis H6 was created to address this. See lines 261 to 275. 
  8. "The chosen metrics cover a part of the process but in order to have an exhaustiveanalysisthe used platform should have been taken into consideration when it comes to interface and the degree of use. Usually, the used platform has a big impact on engagement and participation, the research could have measured how the platform impacts the engagement, this could have been an important metric, and it would’ve complete the research and it would’ve opened a new comparison topic and in that way the researched impact would have been more detailed depending on the platform": See lines 272 to 275. 
  9. "The crowdsourcing platform is an important part of this study, the References used could have contained more than one source about the crowdsourcing platforms and the subject of the platforms could have been addressed as a theoretical background, the article could have been covered what a crowdsourcing platform is and how it is defined in this article. Please see Mindsharing: The Art of Crowdsourcing Everything by Zoref, Lior; Gad Allon, Volodymyr Babich (2020) Crowdsourcing and Crowdfunding in the Manufacturing and Services Sectors. Manufacturing & Service Operations Management 22(1):102-112. https://doi.org/10.1287/msom.2019.0825" : Lines 43 to 49. 
  10. "The whole crowdsourcing process is based on mind-sharing, besides the fact that the article should have taken into consideration that concept, the References would have been more complete with at least a mind-sharing source": See lines 56-82. 

I trust that the amendments which have been made address your comments and I believe that the study became more complete following the amendments. I thank you once more for your review. 

Sincerely yours, 

The author

Round 2

Reviewer 2 Report

The authors have sufficiently addressed most of the recommendations from the reviewer.

The connection between the crowdsourcing pages and the airline page(s) are still unclear. Maybe you could give an example of how the "workflow" is for the user. 

The section about eye-tracking is still only explained by one episodic example. Thus, the experiment presented in Fig. 12/13 does not give evidence. I suggest that Section 3.4 is removed from the paper, and its content should be future work or part of a separate paper that concentrates on eye-tracking and results from the experiment. This would make the goals of the paper clearer.

If you choose to keep this section, the experiment must be better explained, for example, how were the participants selected, how were the tasks in more detail as currently explained, how was the evaluation done (it seems that some tool is used; this tool is not further explained); how the observation that is presented in Fig 12/13 can be generalised (not all participants will concentrate as explained), etc.

Figures 4 to 11: not all values from the legend are visible in the graphs. Some values, such as "unique visitors" in Fig 4 and Fig 5 are not explained; why is this value constant? Further, what type of chart do you present? For example, are the values in Fig. 5 added or separately presented? E.g., at t=78, is the value 1.5''' only for the organic traffic, or the sum of organic traffic and paid traffic? Fig 6: what is the value for "unique visitors"? Is it similar to the paid keywords airlines value or is it constant? Further, is "crowdsourcing organic traffic" a low constant value? And so on.

What is the metric for the Global Rank? (number of bookings, revenue, other financial values, number of flights, web pages access, etc. etc. ?) Global Rank is not explained in Table 1.

line 79: something is wrong: "... on the goal of "

line 82: --> "... has many benefits for entrepreneurs [10]."

line 228: Hypothesis H1 with capital "H".

line 261: H6 is formulated as a question, not as a hypothesis. Please remove "Do" and the question mark.

line 277: H7 is formulated as a question, not as a hypothesis. Please rewrite. (Please note that I would recommend that H7 is removed from this paper, and be part of a separate paper).

Further: "authentication" is probably not what you want to express. I would recommend "evaluation" or "verification".

Line 285: Instead of "This can be achieved", it should be expressed that your study evaluates this ... (using "This can be achieved", the number of participants (8) would be a relevant number; however, the number is a choice you took in your study).

line 502: -> on the programming ...

line 515: Section with capital S

line 512ff: complicated and long sentence.

line 612: "In respect of ..." -> "Regarding ..."

Author Response

Dear Sir/Madam,

Thank you for your recommendations. Below please find the responses to your recommendations:

  1. The connection between the crowdsourcing pages and the airline page(s) are still unclear. Maybe you could give an example of how the "workflow" is for the user”: An example has been provided at lines 172 to 182. 
  2. The section about eye-tracking is still only explained by one episodic example. Thus, the experiment presented in Fig. 12/13 does not give evidence. I suggest that Section 3.4 is removed from the paper, and its content should be future work or part of a separate paper that concentrates on eye-tracking and results from the experiment. This would make the goals of the paper clearer”: The eye-tracking analysis has been removed from the entirety of the paper.
  3. If you choose to keep this section, the experiment must be better explained, for example, (1) how were the participants selected,(2) how were the tasks in more detail as currently explained, (3) how was the evaluation done (it seems that some tool is used; this tool is not further explained); (4) how the observation that is presented in Fig 12/13 can be generalised (not all participants will concentrate as explained)”: The eye-tracking analysis, including section 3.4 has been removed from the entirety of the paper.
  4. "Figures 4 to 11: not all values from the legend are visible in the graphs. Some values, such as "unique visitors" in Fig 4 and Fig 5 are not explained; why is this value constant?. Further, what type of chart do you present?. For example, are the values in Fig. 5 added or separately presented?. g., at t=78, is the value 1.5''' only for the organic traffic, or the sum of organic traffic and paid traffic?. Fig 6: what is the value for "unique visitors"?": Responses can be found at the following lines: 862-870, 846-848 and 903-908.
  5. What is the metric for the Global Rank? (number of bookings, revenue, other financial values, number of flights, web pages access, etc. etc. ?) Global Rank is not explained in Table 1”: See table 1 line 505 under the metric “Global Rank”.
  6. line 79: something is wrong: "... on the goal of ": Done
  7. line 82: --> "... has many benefits for entrepreneurs [10].": This has been removed.
  8. Line 228: Hypothesis H1 with capital "H": Done
  9. line 261: H6 is formulated as a question, not as a hypothesis. Please remove "Do" and the question mark”: Done
  10. line 277: H7 is formulated as a question, not as a hypothesis. Please rewrite. (Please note that I would recommend that H7 is removed from this paper, and be part of a separate paper)”: H7 has been removed entirely.
  11. Further: "authentication" is probably not what you want to express. I would recommend "evaluation" or "verification": This has been removed as H7 has been removed from the paper.
  12. Line 285: Instead of "This can be achieved", it should be expressed that your study evaluates this ... (using "This can be achieved", the number of participants (8) would be a relevant number; however, the number is a choice you took in your study)”: This has been removed as the neuromarketing researched has been removed from the paper.
  13. line 502: -> on the programming ...”: Done
  14. Line 515: Section with capital S”: Done
  15. line 512ff: complicated and long sentence”: This has been amended.
  16. line 612: "In respect of ..." -> "Regarding ...": Done

I trust that the above addresses your comments. Once again, thank you for your recommendations. 

Sincerely yours, 

The author 

Reviewer 3 Report

Thank you for the opportunity to review the paper "The impact of organic traffic of Crowdsourcing platforms on Airlines’ website traffic and user engagement". The paper addresses an interesting and novel theme.

The authors responded and corrected all the remarks and observations highlighted and the results suggest a more consistent and logical text with a consistent reference list. To sum it up, the authors developed a more in-depths theoretical presentation about crowdsourcing, integrating the two suggested aspects: participatory culture and mind-sharing and these are also developed in the methodological framework and the hypotheses tested. The Conclusion and Limitations of this study are also modified and the reference list is updated.

In the end, I would like to make some minor adjustments and recommendations:

  • The word `Crowdsourcing` is interchangeably used with capital letter `C` or `c` in the text (see the title, too).
  • Also, the same for `Crowdfunding`.

I consider that the paper is publishable after a final check from the authors. 

Author Response

Dear Sir/Madam,

Thank you for your recommendations.

The suggested amendments have been implemented throughout the paper. The term crowdsourcing appears with a lower letter “c” throughout the paper, except in areas where the term crowdsourcing refers to a metric, in which case, in that instance, crowdsourcing appears with a capital letter “C”.

The term crowdfunding has also been amended to only appear with a lower letter “c” throughout the paper.

I would like to thank you once again for providing your reviews and recommendations for my study.

Sincerely yours,

The author